# Exactness of Approximate MAP Inference in Continuous MRFs

**Nicholas Ruozzi**
Department of Computer Science
University of Texas at Dallas
Richardson, TX 75080

## Abstract

Computing the MAP assignment in graphical models is generally intractable. As a result, for discrete graphical models, the MAP problem is often approximated using linear programming relaxations. Much research has focused on characterizing when these LP relaxations are tight, and while they are relatively well-understood in the discrete case, only a few results are known for their continuous analog. In this work, we use graph covers to provide necessary and sufficient conditions for continuous MAP relaxations to be tight. We use this characterization to give simple proofs that the relaxation is tight for log-concave decomposable and log-supermodular decomposable models. We conclude by exploring the relationship between these two seemingly distinct classes of functions and providing specific conditions under which the MAP relaxation can and cannot be tight.

## 1 Introduction

Graphical models are a popular modeling tool for both discrete and continuous distributions. We are commonly interested in one of two inference tasks in graphical models: finding the most probable assignment (a.k.a., MAP inference) and computing marginal distributions. These problems are NP-hard in general, and a variety of approximate inference schemes are used in practice.

In this work, we will focus on approximate MAP inference. For discrete state spaces, linear programming relaxations of the MAP problem (specifically, the MAP LP) are quite common [1; 2; 3]. These relaxations replace global marginalization constraints with a collection of local marginalization constraints. Wald and Globerson [4] refer to these as local consistency relaxations (LCRs). The advantage of LCRs is that they are often much easier to specify and to optimize over (e.g., by using a message-passing algorithm such as loopy belief propagation (LBP)). However, the analogous relaxations for continuous state spaces may not be compactly specified and can lead to an unbounded number of constraints (except in certain special cases). To overcome this problem, further relaxations have been proposed [5; 4]. By construction, each of these further relaxations can only be tight if the initial LCR was tight. As a result, there are compelling theoretical and algorithmic reasons to investigate when LCRs are tight.

Among the most well-studied continuous models are the Gaussian graphical models. For this class of models, it is known that the continuous MAP relaxation is tight when the corresponding inverse covariance matrix is positive definite and scaled diagonally dominant (a special case of the so-called log-concave decomposable models)[4; 6; 7]. In addition, LBP is known to converge to the correct solution for Gaussian graphical models and log-concave decomposable models that satisfy a scaled diagonal dominance condition [8; 9]. While much of the prior work in this domain has focused on log-concave graphical models, in this work, we provide a general necessary and sufficient condition for the continuous MAP relaxation to be tight. This condition mirrors the known results for the discrete case and is based on the notion of graph covers: the MAP LP is tight if and only if the

optimal solution to the MAP problem is an upper bound on the MAP solution over any graph cover, appropriately scaled. This characterization will allow us to understand when the MAP relaxation is tight for more general models.

Apart from this characterization theorem, the primary goal of this work is to move towards a uniform treatment of the discrete and continuous cases; they are not as different as they may initially appear. To this end, we explore the relationship between log-concave decomposable models and log-supermodular decomposable models (introduced here in the continuous case). Log-supermodular models provide an example of continuous graphical models for which the MAP relaxation is tight, but the objective function is not necessarily log-concave. These two concepts have analogs in discrete state spaces. In particular, log-concave decomposability is related to log-concave closures of discrete functions and log-supermodular decomposability is a known condition which guarantees that the MAP LP is exact in the discrete setting. We prove a number of results that highlight the similarities and differences between these two concepts as well as a general condition under which the MAP relaxation corresponding to a pairwise twice continuously differentiable model cannot be tight.

## 2 Prerequisites

Let $f : \mathcal{X}^n \to \mathbb{R}_{\geq 0}$ be a non-negative function where $\mathcal{X}$ is the set of possible assignments of each variable. A function $f$ factors with respect to a hypergraph $G = (V, \mathcal{A})$, if there exist potential functions $f_i : \mathcal{X} \to \mathbb{R}_{\geq 0}$ for each $i \in V$ and $f_\alpha : \mathcal{X}^{|\alpha|} \to \mathbb{R}_{\geq 0}$ for each $\alpha \in \mathcal{A}$ such that

$$f(x_1, \ldots, x_n) = \prod_{i \in V} f_i(x_i) \prod_{\alpha \in \mathcal{A}} f_\alpha(x_\alpha).$$

The hypergraph $G$ together with the potential functions $f_{i \in V}$ and $f_{\alpha \in \mathcal{A}}$ define a graphical model.

We are interested computing $\sup_{x \in \mathcal{X}^n} f^G(x)$. In general, this MAP inference task is NP-hard, but in practice, local message-passing algorithms based on approximations from statistical physics, such as LBP, produce reasonable estimates in many settings. Much effort has been invested into understanding when LBP solves the MAP problem. In this section, we briefly review approximate MAP inference in the discrete setting (i.e., when $\mathcal{X}$ is a finite set). For simplicity and consistency, we will focus on log-linear models as in [4]. Given a vector of sufficient statistics $\phi_i(x_i) \in \mathbb{R}^k$ for each $i \in V$ and $x_i \in \mathcal{X}$ and a parameter vector $\theta_i \in \mathbb{R}^k$, we will assume that $f_i(x_i) = \exp\left(\langle \theta_i, \phi_i(x_i) \rangle\right)$. Similarly, given a vector of sufficient statistics $\phi_\alpha(x_\alpha)$ for each $\alpha \in \mathcal{A}$ and $x_\alpha \in \mathcal{X}^{|\alpha|}$ and a parameter vector $\theta_\alpha$, we will assume that $f_\alpha(x_\alpha) = \exp\left(\langle \theta_\alpha, \phi_\alpha(x_\alpha) \rangle\right)$. We will write $\phi(x)$ to represent the concatenation of the individual sufficient statistics and $\theta$ to represent the concatenation of the parameters. The objective function can then be expressed as $f^G(x) = \exp\left(\langle \theta, \phi(x) \rangle\right)$.

### 2.1 The MAP LP relaxation

The MAP problem can be formulated in terms of *mean parameters* [10].

$$\sup_{x \in \mathcal{X}^n} \log f(x) = \sup_{\mu \in \mathcal{M}} \langle \theta, \mu \rangle$$
$$\mathcal{M} = \{\mu \in \mathbb{R}^m : \exists \tau \in \Delta \text{ s.t. } \mathbb{E}_\tau[\phi(x)] = \mu\}$$

where $\Delta$ is the space of all densities over $\mathcal{X}^n$ and $\mathcal{M}$ is the set of all realizable mean parameters.

In general, $\mathcal{M}$ is a difficult object to compactly describe and to optimize over. As a result, one typically constructs convex outerbounds on $\mathcal{M}$ that are more manageable. In the case that $\mathcal{X}$ is finite, one such outerbound is given by the MAP LP. For each $i \in V$ and $k \in \mathcal{X}$, define $\phi_i(x_i)_k \triangleq 1_{\{x_i = k\}}$. Similarly, for each $\alpha \in \mathcal{A}$ and $k \in \mathcal{X}^{|\alpha|}$, define $\phi_\alpha(x_\alpha)_k \triangleq 1_{\{x_\alpha = k\}}$. With this choice of sufficient statistics, $\mathcal{M}$ is equivalent to the set of all marginal distributions over the individual variables and elements of $\mathcal{A}$ that arise from some joint probability distribution. The MAP LP is obtained by replacing $\mathcal{M}$ with a relaxation that only enforces local consistency constraints.

$$\mathcal{M}_L = \left\{ \mu \geq 0 : \begin{array}{ll} \sum_{x_{\alpha \setminus \{i\}}} \mu_\alpha(x_\alpha) = \mu_i(x_i), & \text{for all } \alpha \in \mathcal{A}, i \in \alpha, x_i \in \mathcal{X} \\ \sum_{x_i} \mu_i(x_i) = 1, & \text{for all } i \in V \end{array} \right\}$$

The set of constraints, $\mathcal{M}_L$, is known as the local marginal polytope. The approximate MAP problem is then to compute $\max_{\mu \in \mathcal{M}_L} \langle \theta, \mu \rangle$.

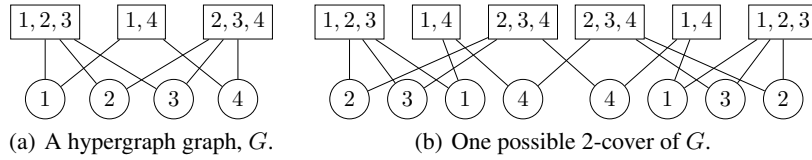

(a) A hypergraph graph, $G$.    (b) One possible 2-cover of $G$.

Figure 1: An example of a graph cover of a factor graph. The nodes in the cover are labeled for the node that they copy in the base graph.

## 2.2 Graph covers

In this work, we are interested in understanding when this relaxation is tight (i.e., when does $\sup_{\mu \in \mathcal{M}_L} \langle \theta, \mu \rangle = \sup_{x \in \mathcal{X}^n} \log f(x)$). For discrete MRFs, the MAP LP is known to be tight in a variety of different settings [11; 12; 13; 14]. Two different theoretical tools are often used to investigate the tightness of the MAP LP: duality and graph covers. Duality has been particularly useful in the design of convergent and correct message-passing schemes that solve the MAP LP [1; 15; 2; 16]. Graph covers provide a theoretical framework for understanding when and why message-passing algorithms such as belief propagation fail to solve the MAP problem [17; 18; 3].

**Definition 2.1.** *A graph $H$ **covers** a graph $G = (V, E)$ if there exists a graph homomorphism $h : H \to G$ such that for all vertices $i \in G$ and all $j \in h^{-1}(i)$, $h$ maps the neighborhood $\partial j$ of $j$ in $H$ bijectively to the neighborhood $\partial i$ of $i$ in $G$.*

If a graph $H$ covers a graph $G$, then $H$ looks locally the same as $G$. In particular, local message-passing algorithms such as LBP have difficulty distinguishing a graph and its covers. If $h(j) = i$, then we say that $j \in H$ is a copy of $i \in G$. Further, $H$ is said to be an $M$-cover of $G$ if every vertex of $G$ has exactly $M$ copies in $H$.

This definition can be easily extended to hypergraphs. Each hypergraph $G$ can be represented in factor graph form: create a node in the factor graph for each vertex (called variable nodes) and each hyperedge (called factor nodes) of $G$. Each factor node is connected via an edge in the factor graph to the variable nodes on which the corresponding hyperedge depends. For an example of a 2-cover, see Figure 1.

To any $M$-cover $H = (V^H, \mathcal{A}^H)$ of $G$ given by the homomorphism $h$, we can associate a collection of potentials: the potential at node $i \in V^H$ is equal to $f_{h(i)}$, the potential at node $h(i) \in G$, and for each $\beta \in \mathcal{A}^H$, we associate the potential $f_{h(\beta)}$. In this way, we can construct a function $f^H : \mathcal{X}^{M|V|} \to \mathbb{R}_{\geq 0}$ such that $f^H$ factorizes over $H$. We will say that the graphical model $H$ is an $M$-cover of the graphical model $G$ whenever $H$ is an $M$-cover of $G$ and $f^H$ is chosen as described above. It will be convenient in the sequel to write $f^H(x^H) = f^H(x^1, \ldots, x^M)$ where $x_i^m$ is the $m^{\text{th}}$ copy of variable $i \in V$.

There is a direct correspondence between $\mu \in \mathcal{M}_L$ and assignments on graph covers. This correspondence is the basis of the following theorem.

**Theorem 2.2** (Ruozzi and Tatikonda 3)**.**

$$\sup_{\mu \in \mathcal{M}_L} \langle \theta, \mu \rangle = \sup_M \sup_{H \in \mathcal{C}^M(G)} \sup_{x^H} \frac{1}{M} \log f^H(x^H)$$

*where $\mathcal{C}^M(G)$ is the set of all $M$-covers of $G$.*

Theorem 2.2 claims that the optimal value of the MAP LP is equal to the supremum over all MAP assignments over all graph covers, appropriately scaled. In particular, the proof of this result shows that, under mild conditions, there exists an $M$-cover $H$ of $G$ and an assignment $x^H$ such that $\frac{1}{M} \log f^H(x^H) = \sup_{\mu \in \mathcal{M}_L} \langle \theta, \mu \rangle$.

## 3 Continuous MRFs

In this section, we will describe how to extend the previous results from discrete to continuous MRFs (i.e., $\mathcal{X} = \mathbb{R}$) using graph covers. The relaxation that we consider here is the appropriate extension

of the MAP LP where each of the sums are replaced by integrals [4].

$$\mathcal{M}_L = \left\{ \mu : \begin{array}{ll} \exists \text{ densities } \tau_i, \tau_\alpha \text{ s.t.} & \\ \int \tau_\alpha(x_\alpha) dx_{\alpha\setminus i} = \tau_i(x_i), & \text{for all } \alpha \in \mathcal{A}, i \in \alpha, x_i \in \mathcal{X} \\ \mu_i = \mathbb{E}_{\tau_i}[\phi_i], & \text{for all } i \in V \\ \mu_\alpha = \mathbb{E}_{\tau_\alpha}[\phi_\alpha], & \text{for all } \alpha \in \mathcal{A} \end{array} \right\}$$

Our goal is to understand under what conditions this continuous relaxation is tight. Wald and Globerson [4] have approached this problem by introducing a further relaxation of $\mathcal{M}_L$ which they call the weak local consistency relaxation (weak LCR). They provide conditions under which the weak LCR (and hence the above relaxation) is tight. In particular, they show that weak LCR is tight for the class of log-concave decomposable models. In this work, we take a different approach. We first prove the analog of Theorem 2.2 in the continuous case and then we show that the known conditions that guarantee tightness of the continuous relaxation are simple consequences of this general theorem.

**Theorem 3.1.**

$$\sup_{\mu \in \mathcal{M}_L} \langle \theta, \mu \rangle = \sup_M \sup_{H \in \mathcal{C}^M(G)} \sup_{x^H} \frac{1}{M} \log f^H(x^H)$$

*where $\mathcal{C}^M(G)$ is the set of all $M$-covers of $G$.*

The proof of Theorem 3.1 is conceptually straightforward, albeit technical, and can be found in Appendix A. The proof approximates the expectations in $\mathcal{M}_L$ as expectations with respect to simple functions, applies the known results for finite spaces, and takes the appropriate limit. Like its discrete counterpart, Theorem 3.1 provides necessary and sufficient conditions for the continuous relaxation to be tight. In particular, for the relaxation to be tight, the optimal solution on any $M$-cover, appropriately scaled, cannot exceed the value of the optimal solution of the MAP problem over $G$.

## 3.1 Tightness of the MAP relaxation

Theorem 3.1 provides necessary and sufficient conditions for the tightness of the continuous relaxation. However, checking that the maximum value attained on any $M$-cover is bounded by the maximum value over the base graph to the $M$, in and of itself, appears to be a daunting task. In this section, we describe two families of graphical models for which this condition is easy to verify: the log-concave decomposable functions and the log-supermodular decomposable functions. Log-concave decomposability has been studied before, particularly in the case of Gaussian graphical models. Log-supermodularity with respect to graphical models, however, appears to have been primarily studied in the discrete case.

### 3.1.1 Log-concave decomposability

A function $f : \mathbb{R}^n \to \mathbb{R}_{\geq 0}$ is log-concave if $f(x)^\lambda f(y)^{1-\lambda} \leq f(\lambda x + (1 - \lambda)y)$ for all $x, y \in \mathbb{R}^n$ and all $\lambda \in [0, 1]$. If $f$ can be written as a product of log-concave potentials over a hypergraph $G$, we say that $f$ is log-concave decomposable over $G$.

**Theorem 3.2.** *If $f$ is log-concave decomposable, then $\sup_x \log f(x) = \sup_{\mu \in \mathcal{M}_L} \langle \theta, \mu \rangle$.*

*Proof.* By log-concave decomposability, for any $M$-cover $H$ of $G$,

$$f^H(x^1, \ldots, x^M) \leq f^G \left( \frac{x^1 + \cdots + x^M}{M} \right)^M,$$

which we obtain by applying the definition of log-concavity separately to each of the $M$ copies of the potential functions for each node and factor of $G$. As a result, $\sup_{x^1, \ldots, x^M} f^H(x^1, \ldots, x^M) \leq \sup_x f^G(x)^M$. The proof of the theorem then follows by applying Theorem 3.1. $\square$

Wald and Globerson [4] provide a different proof of Theorem 3.2 by exploiting duality and the weak LCR.

### 3.1.2 Log-supermodular decomposability

Log-supermodular functions have played an important role in the study of discrete graphical models, and log-supermodularity arises in a number of classical correlations inequalities (e.g., the FKG inequality). For log-supermodular decomposable models, the MAP LP is tight and the MAP problem can be solved exactly in polynomial time [19; 20]. In the continuous case, log-supermodularity is defined analogously to the discrete case. That is, $f : \mathbb{R}^n \to \mathbb{R}_{\geq 0}$ is log-supermodular if $f(x)f(y) \leq f(x \wedge y)f(x \vee y)$ for all $x, y \in \mathbb{R}^n$, where $x \vee y$ is the componentwise maximum of the vectors $x$ and $y$ and $x \wedge y$ is the componentwise minimum. Continuous log-supermodular functions are sometimes said to be multivariate totally positive of order two [21]. We will say that a graphical model is log-supermodular decomposable if $f$ can be factorized as a product of log-supermodular potentials.

For any collection of vectors $x^1, \dots, x^k \in \mathbb{R}^n$, let $z^i(x^1, \dots, x^k)$ be the vector whose $j^{th}$ component is the $i^{th}$ largest element of $x_j^1, \dots, x_j^k$ for each $j \in \{1, \dots, n\}$.

**Theorem 3.3.** *If $f$ is log-supermodular decomposable, then $\sup_x \log f(x) = \sup_{\mu \in \mathcal{M}_L} \langle \theta, \mu \rangle$.*

*Proof.* By log-supermodular decomposability, for any $M$-cover $H$ of $G$,

$$f^H(x^1, \dots, x^M) \leq \prod_{m=1}^{M} f^G(z^m(x^1, \dots, x^M)).$$

Again, this follows by repeatedly applying the definition of log-supermodularity separately to each of the $M$ copies of the potential functions for each node and factor of $G$. As a result, $\sup_{x^1, \dots, x^M} f^H(x^1, \dots, x^M) \leq \sup_{x^1, \dots, x^M} \prod_{m=1}^{M} f^G(x^m)$. The proof of the theorem then follows by applying Theorem 3.1. $\square$

## 4 Log-supermodular decomposability vs. log-concave decomposability

As discussed above, log-concave decomposable and log-supermodular decomposable models are both examples of continuous graphical models for which the MAP relaxation is tight. These two classes are not equivalent: twice continuously differentiable functions are supermodular if and only if all off diagonal elements of the Hessian matrix are non-negative. Contrast this with twice continuously differentiable concave functions where the Hessian matrix must be negative semidefinite. In particular, this means that log-supermodular functions can be multimodel. In this section, we explore the relationship between log-supermodularity and log-concavity.

### 4.1 Gaussian MRFs

We begin with the case of Gaussian graphical models, i.e., pairwise graphical models given by

$$f(x) \propto = \left(-1/2 x^T A x + b^T x\right) = \prod_{i \in V} \exp\left(-\frac{1}{2} A_{ii} x_i^2 + b_i x_i\right) \prod_{(i,j) \in E} \exp\left(-A_{ij} x_i x_j\right)$$

for some symmetric positive definite matrix $A \in \mathbb{R}^{n \times n}$ and vector $b \in \mathbb{R}^n$. Here, $f$ factors over the graph $G$ corresponding to the non-zero entries of the matrix $A$.

Gaussian graphical models are a relatively well-studied class of continuous graphical models. In fact, sufficient conditions for the convergence and correctness of Gaussian belief propagation (GaBP) are known for these models. Specifically, GaBP converges to the optimal solution if the positive definite matrix $A$ is walk-summable, scaled diagonally dominant, or log-concave decomposable [22; 7; 8; 9]. These conditions are known to be equivalent [23; 6].

**Definition 4.1.** $\Gamma \in \mathbb{R}^{n \times n}$ is **scaled diagonally dominant** if $\exists w \in \mathbb{R}^n, w > 0$ such that $|\Gamma_{ii}|w_i > \sum_{j \neq i} |\Gamma_{ij}|w_j$.

In addition, the following theorem provides a characterization of scaled diagonal dominance (and hence log-concave decomposability) in terms of graph covers for these models.

**Theorem 4.2** (Ruozzi and Tatikonda 6)**.** *Let $A$ be a symmetric positive definite matrix. The following are equivalent.*

1. *A is scaled diagonally dominant.*

2. *All covers of A are positive definite.*

3. *All 2-covers of A are positive definite.*

The proof of this theorem constructs a specific 2-cover whose covariance matrix has negative eigenvalues whenever the matrix $A$ is positive definite but not scaled diagonally dominant. The joint distribution corresponding to this 2-cover is not bounded from above, so the optimal value of the MAP relaxation is $+\infty$ as per Theorem 3.1.

For Gaussian graphical models, log-concave decomposability and log-supermodular decomposability are related: every positive definite, log-supermodular decomposable model is log-concave decomposable, and every positive definite, log-concave decomposable model is a signed version of some positive definite, log-supermodular decomposable Gaussian graphical model. This follows from the following simple lemma.

**Lemma 4.3.** *A symmetric positive definite matrix A is scaled diagonally dominant if and only if the matrix B such that $B_{ii} = A_{ii}$ for all $i$ and $B_{ij} = -|A_{ij}|$ for all $i \neq j$ is positive definite.*

If $A$ is positive definite and scaled diagonally dominant, then the model is log-concave decomposable. In contrast, the model would be log-supermodular decomposable if all of the off-diagonal elements of $A$ were negative, independent of the diagonal. In particular, the diagonal could have both positive and negative elements, meaning that $f(x)$ could be either log-concave, log-convex, or neither. As log-convex quadratic forms do not correspond to normalizable Gaussian graphical models, the log-convex case appears to be less interesting as the MAP problem is unbounded from above. However, the situation is entirely different for constrained (over some convex set) log-quadratic maximization. As an example, consider a box constrained log-quadratic maximization problem where the matrix $A$ has all negative off-diagonal entries. Such a model is always log-supermodular decomposable. Hence, the MAP relaxation is tight, but the model is not necessarily log-concave.

## 4.2 Pairwise twice differentiable MRFs

All of the results from the previous section can be extended to general twice continuously differentiable functions over pairwise graphical models (i.e., $|\alpha| = 2$ for all $\alpha \in \mathcal{A}$). In this section, unless otherwise specified, assume that all models are pairwise.

**Theorem 4.4.** *If $\log f(x)$ is strictly concave and twice continuously differentiable, the following are equivalent.*

1. $\nabla^2(\log f)(x)$ *is scaled diagonally dominant for all $x$.*

2. $\nabla^2(\log f^H)(x^H)$ *is negative definite for every graph cover $H$ of $G$ and every $x^H$.*

3. $\nabla^2(\log f^H)(x^H)$ *is negative definite for every 2-cover $H$ of $G$ and every $x^H$.*

The equivalence of 1-3 in Theorem 4.4 follows from Theorem 4.2.

**Corollary 4.5.** *If $\nabla^2(\log f)(x)$ is scaled diagonally dominant for all $x$, then the continuous MAP relaxation is tight.*

**Corollary 4.6.** *If $f$ is log-concave decomposable over a pairwise graphical model and strictly log-concave, then $\nabla^2(\log f)(x)$ is scaled diagonally dominant for all $x$.*

Whether or not log-concave decomposability is equivalent to the other conditions listed in the statement of Theorem 4.4 remains an open question (though we conjecture that this is the case). Similar ideas can be extended to general twice continuously differentiable functions.

**Theorem 4.7.** *Suppose $\log f(x)$ is twice continuously differentiable with a maximum at $x^*$. Let $B_{ij} = |\nabla^2(\log f)(x^*)_{ij}|$ for all $i \neq j$ and $B_{ii} = \nabla^2(\log f)(x^*)_{ii}$. If $f$ admits a pairwise factorization over $G$ and $B$ has both positive and negative eigenvalues, then the continuous MAP relaxation is **not** tight.*

*Proof.* If $B$ has both positive and negative eigenvalues, then there exists a 2-cover $H$ of $G$ such that $\nabla^2(\log f^H)(x^*, x^*)$ has both positive and negative eigenvalues. As a result, the lift of $x^*$ to the

2-cover $f^H$ is a saddle point. Consequently, $f^H(x^*, x^*) < \sup_{x^H} f^H(x^H)$. By Theorem 3.1, the continuous MAP relaxation cannot be tight. □

This negative result is quite general. If $\nabla^2(\log f)$ is positive definite but not scaled diagonally dominant at any global optimum, then the MAP relaxation is not tight. In particular, this means that all log-supermodular decomposable functions that meet the conditions of the theorem must be s.d.d. at their optima.

Algorithmically, Moallemi and Van Roy [9] argued that belief propagation converges for models that are log-concave decomposable and scaled diagonally dominant. It is unknown whether or not a similar convergence argument applies to log-supermodular decomposable functions.

### 4.3   Concave closures

Many of the tightness results in the discrete case can be seen as a specific case of the continuous results described above. Again, suppose that $\mathcal{X} \subset \mathbb{R}$ is a finite set.

**Definition 4.8.** *The concave closure of a function $g : \mathcal{X}^n \to \mathbb{R} \cup \{-\infty\}$ at $x \in \mathbb{R}^n$ is given by*

$$\overline{g}(x) = \sup \left\{ \sum_{y \in \mathcal{X}^n} \lambda(y) g(y) : \textstyle\sum_y \lambda(y) = 1, \sum_y \lambda(y) y = x, \lambda(y) \geq 0 \right\}$$

Equivalently, the concave closure of a function is the smallest concave function such that $g(x) \leq \overline{g}(x)$ for all $x$. A function and its concave closure must necessarily have the same maximum. Computing the concave (or convex) closure of a function is NP-hard in general, but it can be efficiently computed for certain special classes of discrete functions. In particular, when $\mathcal{X} = \{0, 1\}$ and $\log f$ is supermodular, then its concave closure can be computed in polynomial time as it is equal to the Lovász extension of $\log f$. The Lovász extension has a number of interesting properties. Most notably, it is linear (the Lovász extension of a sum of functions is equal to sum of the Lovász extensions). Define the log-concave closure of $f$ to be $\hat{f}(x) = \exp(\overline{\log f}(x))$. As a result, if $f$ is log-supermodular decomposable, then $\hat{f}$ is log-concave decomposable.

**Theorem 4.9.** *If $\hat{f} = \prod_{i \in V} \hat{f}_i \prod_{\alpha \in \mathcal{A}} \hat{f}_\alpha$, then $\sup_{x \in \mathcal{X}^n} f(x) = \sum_{\mu \in \mathcal{M}_L} \langle \theta, \mu \rangle$.*

This theorem is a direct consequence of Theorem 3.2. For example, the tightness results of Bayati et al. [11] and Sanghavi et al. [14] (and indeed many others) can be seen as a special case of this theorem. Even when $|\mathcal{X}|$ is not finite, the concave closure can be similarly defined, and the theorem holds in this case as well. Given the characterization in the discrete case, this suggests that there could be a, possibly deep, connection between log-concave closures and log-supermodular decomposability.

## 5   Discussion

We have demonstrated that the same necessary and sufficient condition based on graph covers for the tightness of the MAP LP in the discrete case translates seamlessly to the continuous case. This characterization allowed us to provide simple proofs of the tightness of the MAP relaxation for log-concave decomposable and log-supermodular decomposable models. While the proof of Theorem 3.1 is nontrivial, it provides a powerful tool to reason about the tightness of MAP relaxations. We also explored the intricate relationship between log-concave and log-supermodular decomposablity in both the discrete and continuous cases which provided intuition about when the MAP relaxation can or cannot be tight for pairwise graphical models.

## A   Proof of Theorem 3.1

The proof of this theorem proceeds in two parts. First, we will argue that

$$\sup_{\mu \in \mathcal{M}_L} \langle \theta, \mu \rangle \geq \sup_M \sup_{H \in \mathcal{C}^M(G)} \sup_{x^H} \frac{1}{M} \log f^H(x^H).$$

To see this, fix an $M$-cover, $H$, of $G$ via the homomorphism $h$ and consider any assignment $x^H$. Construct the mean parameters $\mu' \in \mathcal{M}_L$ as follows.

$$\tau_i(x_i) = \frac{1}{M} \sum_{j \in V(H):h(j)=i} \delta(x_j^H - x_i) \qquad \mu_i' = \int \tau_i(x_i)\phi_i(x_i)dx_i$$

$$\tau_\alpha(x_\alpha) = \frac{1}{M} \sum_{\beta \in \mathcal{A}(H):h(\beta)=\alpha} \delta(x_\beta^H - x_\alpha) \qquad \mu_\alpha' = \int \tau_\alpha(x_\alpha)\phi_\alpha(x_\alpha)dx_\alpha$$

Here, $\delta(\cdot)$ is the Dirac delta function[1]. This implies that

$$\frac{1}{M} \log f^H(x^H) = \langle \theta, \mu' \rangle \leq \sup_{\mu \in \mathcal{M}_L} \langle \theta, \mu \rangle.$$

For the other direction, fix some $\mu' \in \mathcal{M}_L$ such that $\mu'$ is generated by the vector of densities $\tau$. We will prove the result for locally consistent probability distributions with bounded support. The result for arbitrary $\tau$ will then follow by constructing sequences of these distributions that converge (in measure) to $\tau$. For simplicity, we will assume that each potential function is strictly positive[2].

Consider the space $[-t, t]^{|V|}$ for some positive integer $t$. We will consider local probability distributions that are supported on subsets of this space. That is, $\text{supp}(\tau_i) \subseteq [-t, t]$ for each $i$ and $\text{supp}(\tau_\alpha) \subseteq [-t, t]^{|\alpha|}$ for each $\alpha$. For a fixed positive integer $s$, divide the interval $[-t, t]$ into $2^{s+1}t$ intervals of size $1/2^s$ and let $\mathcal{S}_k$ denote the $k^{\text{th}}$ interval. This partitioning divides $[-t, t]^{|V|}$ into disjoint cubes of volume $1/2^{s|V|}$. The distribution $\tau$ can be approximated as a sequence of distributions $\tau^1, \tau^2, \dots$ as follows. Define a vector of approximate densities $\tau^s$ by setting

$$\tau_i^s(x_i') \triangleq \begin{cases} 2^s \int_{\mathcal{S}_k} \tau_i(x_i)dx_i, & \text{if } x_i' \in \mathcal{S}_k \\ 0, & \text{otherwise} \end{cases}$$

$$\tau_\alpha^s(x_\alpha') \triangleq \begin{cases} 2^{|\alpha|s} \int_{\prod_{k_j:j \in \alpha} \mathcal{S}_{k_j}} \tau_\alpha(x_\alpha)dx_\alpha, & \text{if } x_\alpha' \in \prod_{k_j:j \in \alpha} \mathcal{S}_{k_j} \\ 0, & \text{otherwise} \end{cases}$$

We have $\tau^s \to \tau$, $\int_{[-t,t]} \tau_i^s(x_i)\phi_i(x_i)dx_i \to \mu_i'$ for each $i \in V(G)$, and $\int_{[-t,t]^{|\alpha|}} \tau_\alpha^s(x_\alpha)\phi_\alpha(x_\alpha)dx_\alpha \to \mu_\alpha'$ for each $\alpha \in \mathcal{A}(G)$.

The continuous MAP relaxation for local probability distributions of this form can be expressed in terms of discrete variables over $\mathcal{X} = \{1, \dots, 2^{s+1}t\}$. To see this, define $\mu_i^s(z_i) = \int_{S_{z_i}} \tau_i^s(x_i)dx_i$ for each $z_i \in \{1, \dots, 2^{s+1}t\}$ and $\mu_\alpha^s(z_\alpha) = \int_{S_{z_\alpha}} \tau_\alpha^s(x_\alpha)dx_\alpha$ for each $z_\alpha \in \{1, \dots, 2^{s+1}t\}^{|\alpha|}$. The corresponding MAP LP objective, evaluated at $\mu^s$, is then

$$\sum_{i \in V} \sum_{z_i} \mu_i^s(z_i) \int_{\mathcal{S}_{z_i}} 2^s \log f_i(x_i)dx_i + \sum_{\alpha \in \mathcal{A}} \sum_{z_\alpha} \mu_\alpha^s(z_\alpha) \int_{\mathcal{S}_{z_\alpha}} 2^{|\alpha|s} \log f_\alpha(x_\alpha)dx_\alpha. \tag{1}$$

This MAP LP objective corresponds to a discrete graphical model that factors over the hypergraph $G$, with potential functions corresponding to the above integrals over the partitions indexed by the vector $z$.

$$g^s(z) \propto \prod_{i \in V(G)} \exp\left(\int_{\mathcal{S}_{z_i}} 2^s \log f_i(x_i)dx_i\right) \prod_{\alpha \in \mathcal{A}(G)} \exp\left(\int_{\mathcal{S}_{z_\alpha}} 2^{|\alpha|s} \log f_\alpha(x_\alpha)dx_\alpha\right)$$

$$= \prod_{i \in V(G)} \exp\left(\int_{\mathcal{S}_z} 2^{|V(G)|s} \log f_i(x_i)dx\right) \prod_{\alpha \in \mathcal{A}(G)} \exp\left(\int_{\mathcal{S}_z} 2^{|V(G)|s} \log f_\alpha(x_\alpha)dx\right)$$

Every assignment selects a single cube indexed by $z$. The value of the objective is calculated by averaging $\log f$ over the cube indexed by $z$. As a result, $\max_z g^s(z) \leq \sup_x f(x)$ and for any $M$-cover $H$ of $G$, $\max_{z^{1:M}} g^{H,s}(z^1, \dots, z^M) \leq \sup_{x^{1:m}} f^H(x^1, \dots, x^M)$. As this upper bound holds for any fixed $s$, it must also hold for any vector of distributions that can be written as a limit of such distributions. Now, by applying Theorem 2.2 for the discrete case, $\langle \theta, \mu' \rangle = \lim_{s \to \infty} \langle \theta, \mu^s \rangle \leq \sup_M \sup_{H \in \mathcal{C}^M(G)} \sup_{x^H} \frac{1}{M} \log f^H(x^H)$ as desired. To finish the proof, observe that any Riemann integrable density can be arbitrarily well approximated by densities of this form as $t \to \infty$.

## Footnotes

[1] In order to make this precise, we would need to use Lebesgue integration or take a sequence of probability distributions over the space $\mathbb{R}^{M|V|}$ that arbitrarily well-approximate the desired assignment $x^H$.

[2] The same argument will apply in the general case, but each of the local distributions must be contained in the support of the corresponding potential function (i.e., $\text{supp}(\tau_i) \subseteq \text{supp}(f_i)$) for the integrals to exist.

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
