[Reviews · NeurIPS 2015]

Submitted by Assigned_Reviewer_1

The paper is well written.

line 120: missing \ln before f(x) line 196: "has be" -> "has been" line 254: \propto -> = line 291: To talk about Gaussian and non psd A just doesn't make sense. The Gaussian can't be normalized otherwise. If you want to talk about truncated extended Gaussian, then explicitly say so.

In general, one should be more careful about the space of natural parameter when defining continuous MRF, as not all natural parameter in the full R^n is guaranteed to be properly normalized (i.e., the partition function diverges).
Summary: The paper addressed the question of when a local MAP relaxation of a continuous MRF is exact. The authors showed that the graph cover device used to study the same problem in the discrete case pretty much generalizes to the continuous case as well. Based on this, a different proof for a known result (log-concave potential) is provided, and a new sufficient condition (log-supermodularity) is given. Overall this is a decent paper that adds to our understanding of continuous MRF. The downside is that is is not clear if this new intuition would lead to actual computational procedures.

Submitted by Assigned_Reviewer_2

Graph cover is a way to construct relaxations of MAP inference, in which nodes and cliques are copied allowing different assignments to different copies of one variable. It has been established in (Ruozzi and Tatikonda 13) that graph cover (M-cover in particular) is closely related to LP relaxations: the latter is equivalent to the loosest possible M-cover relaxation---LP relaxation is tight if all possible M-covers are tight.

A direct application of the result is to show that log-concave decomposable and log-supermodular decomposable models have tight LP relaxations. The log-concave case has been proved by a different approach in [2], however the prove in this paper is a lot simpler. The result is also applied to analyzing Gaussian MRFs and pairwise twice differentiable MRFs. The analysis also established close relations between discrete and continuous MAP inference problems.

Overall the paper is very well organized and written.
Summary: This paper extends the results of (Ruozzi and Tatikonda 13) characterizing LP relaxation of MAP inference using graph covers from discrete to continuous cases.

The result is then applied to analyzing log-supermodular decomposable and log-concave decomposable models. All these seem to be solid contributions to understanding in this domain.

Submitted by Assigned_Reviewer_3

This paper provides a theoretical study of tightness of the LP relaxations for MRFs with continuous variables. The paper provides the following contributions: - Generalization of the results of [13] to the case of continuous MRFs: formulation of the value of the LP relaxation via graph covers (theorem 3.1) - Formulation of the two special cases when the LP relaxation is tight (log-concave decomposable and log-supermodular decomposable models). Proofs are based on theorem 3.1. The paper introduces the family of log-supermodular decomposable functions. - Condition allowing to identify cases when the MAP relaxation is not tight (theorem 4.7). The paper is relatively well written, although some parts are quite hard to follow.

The derivations presented in the paper seem to be correct. I have several comments related to correctness/clarity: 1) Proof of theorem 3.3 is based on the repeated application of the definition of log-supermodularity (line 222) to the l.h.s. of line 233. This definition allows to upper bound the product of function values at different point by product of the function values at the point-wise maximum and minimum. At the same time the r.h.s line 233 contains only point-wise maximum (z^m) and not the point-wise minimums. Is this error critical for the result to hold? 2) Section 4.1 should be explaining relation of the two families of the functions in case of Gaussian MRFs. I could not get the relation based on the provided explanation. Lines 280-282 were not enough for my understanding. 3) I did not understand why section 4.2 was included into section 4, because it seemed to provide a separate result.

The presented results seem to be original. The connections to other existing results are explained although sometimes are not reader-friendly (e.g. lines 354-355).
Summary: This paper provides theoretical analysis of the tightness of the LP relaxation of the MRFs with continuous variables. The paper is relatively clear and the results seem to be novel and significant enough to justify the publication.

Submitted by Assigned_Reviewer_4

The authors provide an alternative characterization of conditions for tight LP MAP relaxations in continuous MRFs as compared to Wald & Globerson. In addition to the previous class of log-concave decomposable models, the authors show that the LP relaxation is tight for the class of log-supermodular decomposable models.

The connection between log-concave and log-supermodular models needs clarification for Gaussian MRFs. For positive definite models log-supermodular decomposability implies log-concave decomposability (L279-L283). Wald and Globerson show that log-concave decomposability is necessary and sufficient for LP tightness in Gaussian models, then the log-supermodular case is a weaker statement. The non-PSD case discussed in the paragraph starting at L287 is uninteresting since the objective is unbounded on the constraint set. The constrained case at the end of this paragraph is confusing since no details are provided.

Some detailed comments below:

* The current title suggests a paper about an approximate MAP inference algorithm for continuous MRFs, not analysis of them. The title does not mention the main point of the paper, e.g. a graph cover analysis of LP tightness for continuous MAP.

Consider changing to reflect this.

* Does the proof in Appendix A assume f(x) is differentiable? Any other assumptions beyond strict positivity?

* The authors should cite the MPLP work of: D. Sontag et al., UAI 2008
Summary: This is a solid and well written paper. Proof of the main result (Thm 3.1) is fairly involved and some details are glossed over, but it appears correct. The manuscript lacks any experimental validation, but makes up for this weakness with strong technical contributions.

Author Feedback
Author rebuttal: Thank you for your thoughtful reviews, typo corrections, and suggested improvements.

We first comment on the case of Gaussian MRFs as several reviewers found this section a bit confusing. In Section 4.1, we consider graphical models that are log-quadratic. While the quadratic term is required to be concave for Gaussian MRFs, we'd like to consider the broader class of models that allow arbitrary quadratic forms. This seems uninteresting as convex quadratic forms will not be bounded from above. However, there exist log-supermodular, log-quadratic forms that are convex and bounded over a constrained region. Consider such a maximization problem for a log-quadratic, log-convex, and log-supermodular MRF over the unit cube. The optimization problem is bounded, and the relaxation is tight for these models (in spite of the fact that they are not log-concave decomposable).

We further discuss the relationship between log-supermodular and log-concave decomposability. From the paper: "Every positive definite, log-supermodular decomposable model is log-concave decomposable, and every positive definite, log-concave decomposable model is a signed version of some positive definite, log-supermodular decomposable Gaussian graphical model." That is, for Gassuan MRFs, log-concave decomposability is equivalent to the positive definiteness of the unsigned matrix |A| (which is log-supermodular, but not necessarily positive definite).

Reviewer 2:

The proof of Theorem 3.3 should be correct as written, though the notation is admittedly not the best. The jth component of z^m(x^1,...,x^M) is the mth largest element among x^1_j,...,x^M_j. So, z^1_j would be the componentwise maximum over x^1_j,...,x^M_j and z^M_j would be the componentwise minimum over x^1_j,...,x^M_j.

Reviewer 3:

Other than nonnegativity, the proof also implicitly requires that the Riemann integrals of the \phi functions exist over all bounded sets (functions that are continuous almost everywhere have this property).

Reviewer 5:

The optimization problems under consideration don't require that the function f(x) is a probability distribution. You are correct that partition functions/normalizing constants may not always exist for continuous CRFs. As long as the integrals of the individual \phi functions exist on compact intervals, the argument still applies (note that the relaxation may not be interesting, i.e., the optimum of the relaxation could be equal to +\infty).